# Multivariate Probabilistic Time Series Forecasting via Conditioned Normalizing Flows

**Kashif Rasul, Abdul-Saboor Sheikh, Ingmar Schuster, Urs Bergmann & Roland Vollgraf**
Zalando Research
Mühlenstraße 25
10243 Berlin
Germany
{kashif.rasul, ingmar.schuster, roland.vollgraf}@zalando.de

## Abstract

Time series forecasting is often fundamental to scientific and engineering problems and enables decision making. With ever increasing data set sizes, a trivial solution to scale up predictions is to assume independence between interacting time series. However, modeling statistical dependencies can improve accuracy and enable analysis of interaction effects. Deep learning methods are well suited for this problem, but multivariate models often assume a simple parametric distribution and do not scale to high dimensions. In this work we model the multivariate temporal dynamics of time series via an autoregressive deep learning model, where the data distribution is represented by a conditioned normalizing flow. This combination retains the power of autoregressive models, such as good performance in extrapolation into the future, with the flexibility of flows as a general purpose high-dimensional distribution model, while remaining computationally tractable. We show that it improves over the state-of-the-art for standard metrics on many real-world data sets with several thousand interacting time-series.

## 1 Introduction

Classical time series forecasting methods such as those in Hyndman & Athanasopoulos (2018) typically provide univariate forecasts and require hand-tuned features to model seasonality and other parameters. Time series models based on recurrent neural networks (RNN), like LSTM (Hochreiter & Schmidhuber, 1997), have become popular methods due to their end-to-end training, the ease of incorporating exogenous covariates, and their automatic feature extraction abilities, which are the hallmarks of deep learning. Forecasting outputs can either be points or probability distributions, in which case the forecasts typically come with uncertainty bounds.

The problem of modeling uncertainties in time series forecasting is of vital importance for assessing how much to trust the predictions for downstream tasks, such as anomaly detection or (business) decision making. Without probabilistic modeling, the importance of the forecast in regions of low noise (small variance around a mean value) versus a scenario with high noise cannot be distinguished. Hence, point estimation models ignore risk stemming from this noise, which would be of particular importance in some contexts such as making (business) decisions.

Finally, individual time series, in many cases, are statistically dependent on each other, and models need the capacity to adapt to this in order to improve forecast accuracy (Tsay, 2014). For example, to model the demand for a retail article, it is important to not only model its sales dependent on its own past sales, but also to take into account the effect of interacting articles, which can lead to *cannibalization* effects in the case of article competition. As another example, consider traffic flow in a network of streets as measured by occupancy sensors. A disruption on one particular street will also ripple to occupancy sensors of nearby streets—a univariate model would arguably not be able to account for these effects.

In this work, we propose end-to-end trainable autoregressive deep learning architectures for *probabilistic forecasting* that explicitly models multivariate time series and their temporal dynamics by employing a normalizing flow, like the Masked Autoregressive Flow (Papamakarios et al., 2017) or

Real NVP (Dinh et al., 2017). These models are able to scale to thousands of interacting time series, we show that they are able to learn ground-truth dependency structure on toy data and we establish new state-of-the-art results on diverse real world data sets by comparing to competitive baselines. Additionally, these methods adapt to a broad class of underlying data distribution on account of using a normalizing flow and our Transformer based model is highly efficient due to the parallel nature of attention layers while training.

The paper first provides some background context in Section 2. We cover related work in Section 3. Section 4 introduces our model and the experiments are detailed in Section 5. We conclude with some discussion in Section 6. The Appendix contains details of the datasets, additional metrics and exploratory plots of forecast intervals as well as details of our model.

## 2 BACKGROUND

### 2.1 DENSITY ESTIMATION VIA NORMALIZING FLOWS

Normalizing flows (Tabak & Turner, 2013; Papamakarios et al., 2019) are mappings from $\mathbb{R}^D$ to $\mathbb{R}^D$ such that densities $p_{\mathcal{X}}$ on the input space $\mathcal{X} = \mathbb{R}^D$ are transformed into some simple distribution $p_{\mathcal{Z}}$ (e.g. an isotropic Gaussian) on the space $\mathcal{Z} = \mathbb{R}^D$. These mappings, $f \colon \mathcal{X} \mapsto \mathcal{Z}$, are composed of a sequence of bijections or invertible functions. Due to the change of variables formula we can express $p_{\mathcal{X}}(\mathbf{x})$ by

$$p_{\mathcal{X}}(\mathbf{x}) = p_{\mathcal{Z}}(\mathbf{z}) \left| \det \left( \frac{\partial f(\mathbf{x})}{\partial \mathbf{x}} \right) \right|,$$

where $\partial f(\mathbf{x})/\partial \mathbf{x}$ is the Jacobian of $f$ at $\mathbf{x}$. Normalizing flows have the property that the inverse $\mathbf{x} = f^{-1}(\mathbf{z})$ is easy to evaluate and computing the Jacobian determinant takes $O(D)$ time.

The bijection introduced by Real NVP (Dinh et al., 2017) called the *coupling layer* satisfies the above two properties. It leaves part of its inputs unchanged and transforms the other part via functions of the un-transformed variables (with superscript denoting the coordinate indices)

$$\begin{cases} \mathbf{y}^{1:d} = \mathbf{x}^{1:d} \\ \mathbf{y}^{d+1:D} = \mathbf{x}^{d+1:D} \odot \exp(s(\mathbf{x}^{1:d})) + t(\mathbf{x}^{1:d}), \end{cases}$$

where $\odot$ is an element wise product, $s()$ is a scaling and $t()$ a translation function from $\mathbb{R}^d \mapsto \mathbb{R}^{D-d}$, given by neural networks. To model a nonlinear density map $f(\mathbf{x})$, a number of coupling layers which map $\mathcal{X} \mapsto \mathcal{Y}_1 \mapsto \cdots \mapsto \mathcal{Y}_{K-1} \mapsto \mathcal{Z}$ are composed together all the while alternating the dimensions which are unchanged and transformed. Via the change of variables formula the probability density function (PDF) of the flow given a data point can be written as

$$\log p_{\mathcal{X}}(\mathbf{x}) = \log p_{\mathcal{Z}}(\mathbf{z}) + \log|\det(\partial \mathbf{z}/\partial \mathbf{x})| = \log p_{\mathcal{Z}}(\mathbf{z}) + \sum_{i=1}^{K} \log|\det(\partial \mathbf{y}_i/\partial \mathbf{y}_{i-1})|. \quad (1)$$

Note that the Jacobian for the Real NVP is a block-triangular matrix and thus the log-determinant of each map simply becomes

$$\log|\det(\partial \mathbf{y}_i/\partial \mathbf{y}_{i-1})| = \log|\exp(\mathtt{sum}(s_i(\mathbf{y}_{i-1}^{1:d})))|, \quad (2)$$

where $\mathtt{sum}()$ is the sum over all the vector elements. This model, parameterized by the weights of the scaling and translation neural networks $\theta$, is then trained via stochastic gradient descent (SGD) on training data points where for each batch $\mathcal{D}$ we maximize the average log likelihood (1) given by

$$\mathcal{L} = \frac{1}{|\mathcal{D}|} \sum_{\mathbf{x} \in \mathcal{D}} \log p_{\mathcal{X}}(\mathbf{x}; \theta).$$

In practice, Batch Normalization (Ioffe & Szegedy, 2015) is applied as a bijection to outputs of successive coupling layers to stabilize the training of normalizing flows. This bijection implements the normalization procedure using a weighted moving average of the layer's mean and standard deviation values, which has to be adapted to either training or inference regimes.

The Real NVP approach can be generalized, resulting in Masked Autoregressive Flows (Papamakarios et al., 2017) (MAF) where the transformation layer is built as an autoregressive neural network in the sense that it takes in some input $\mathbf{x} \in \mathbb{R}^D$ and outputs $\mathbf{y} = (y^1, \ldots, y^D)$ with the requirement that this transformation is invertible and any output $y^i$ *cannot* depend on input with dimension indices $\geq i$, i.e. $\mathbf{x}^{\geq i}$. The Jacobian of this transformation is triangular and thus the Jacobian determinant is tractable. Instead of using a RNN to share parameters across the $D$ dimensions of $\mathbf{x}$ one avoids this sequential computation by using masking, giving the method its name. The inverse however, needed for generating samples, is sequential.

By realizing that the scaling and translation function approximators don't need to be invertible, it is straight-forward to implement conditioning of the PDF $p_{\mathcal{X}}(\mathbf{x}|\mathbf{h})$ on some additional information $\mathbf{h} \in \mathbb{R}^H$: we concatenate $\mathbf{h}$ to the inputs of the scaling and translation function approximators of the coupling layers, i.e. $s(\texttt{concat}(\mathbf{x}^{1:d}, \mathbf{h}))$ and $t(\texttt{concat}(\mathbf{x}^{1:d}, \mathbf{h}))$ which are modified to map $\mathbb{R}^{d+H} \mapsto \mathbb{R}^{D-d}$. Another approach is to add a bias computed from $\mathbf{h}$ to every layer inside the $s$ and $t$ networks as proposed by Korshunova et al. (2018). This does not change the log-determinant of the coupling layers given by (2). More importantly for us, for sequential data, indexed by $t$, we can share parameters across the different conditioners $\mathbf{h}_t$ by using RNNs or Attention in an autoregressive fashion.

For discrete data the distribution has differential entropy of negative infinity, which leads to arbitrary high likelihood when training normalizing flow models, even on test data. To avoid this one can *dequantize* the data, often by adding $\mathrm{Uniform}[0, 1)$ noise to integer-valued data. The log-likelihood of the resulting continuous model is then lower-bounded by the log-likelihood of the discrete one as shown in Theis et al. (2016).

## 2.2 SELF-ATTENTION

The self-attention based Transformer (Vaswani et al., 2017) model has been used for sequence modeling with great success. The multi-head self-attention mechanism enables it to capture both long- and short-term dependencies in time series data. Essentially, the Transformer takes in a sequence $\mathbf{X} = [\mathbf{x}_1, \ldots, \mathbf{x}_T]^\mathsf{T} \in \mathbb{R}^{T \times D}$, and the multi-head self-attention transforms this into $H$ distinct query $\mathbf{Q}_h = \mathbf{X}\mathbf{W}_h^Q$, key $\mathbf{K}_h = \mathbf{X}\mathbf{W}_h^K$ and value $\mathbf{V}_h = \mathbf{X}\mathbf{W}_h^V$ matrices, where the $\mathbf{W}_h^Q$, $\mathbf{W}_h^K$, and $\mathbf{W}_h^V$ are learnable parameters. After these linear projections the scaled dot-product attention computes a sequence of vector outputs via:

$$\mathbf{O}_h = \mathrm{Attention}(\mathbf{Q}_h, \mathbf{K}_h, \mathbf{V}_h) = \texttt{softmax}\left(\frac{\mathbf{Q}_h\mathbf{K}_h^\mathsf{T}}{\sqrt{d_K}} \cdot \mathbf{M}\right)\mathbf{V}_h,$$

where a mask $\mathbf{M}$ can be applied to filter out right-ward attention (or future information leakage) by setting its upper-triangular elements to $-\infty$ and we normalize by $d_K$ the dimension of the $\mathbf{W}_h^K$ matrices. Afterwards, all $H$ outputs $\mathbf{O}_h$ are concatenated and linearly projected again.

One typically uses the Transformer in an encoder-decoder setup, where some warm-up time series is passed through the encoder and the decoder can be used to learn and autoregressively generate outputs.

## 3 RELATED WORK

Related to this work are models that combine normalizing flows with sequential modeling in some way. Transformation Autoregressive Networks (Oliva et al., 2018) which model the density of a multi-variate variable $\mathbf{x} \in \mathbb{R}^D$ as $D$ conditional distributions $\Pi_{i=1}^D p_{\mathcal{X}}(x^i | x^{i-1}, \ldots, x^1)$, where the conditioning is given by a mixture model coming from the state of a RNN, and is then transformed via a bijection. The PixelSNAIL (Chen et al., 2018) method also models the joint as a product of conditional distributions, optionally with some global conditioning, via causal convolutions and self-attention (Vaswani et al., 2017) to capture long-term temporal dependencies. These methods are well suited to modeling high dimensional data like images, however their use in modeling the temporal development of data has only recently been explored for example in VideoFlow (Kumar et al., 2019) in which they model the distribution of the next video frame via a flow where the model outputs the parameters of the flow's base distribution via a ConvNet, whereas our approach will be based on conditioning of the PDF as described above.

Using RNNs for modeling either multivariate or temporal dynamics introduces sequential computational dependencies that are not amenable to parallelization. Despite this, RNNs have been shown to be very effective in modeling sequential dynamics. A recent work in this direction (Hwang et al., 2019) employs bipartite flows with RNNs for temporal conditioning to develop a conditional generative model of multivariate sequential data. The authors use a bidirectional training procedure to learn a generative model of observations that together with the temporal conditioning through a RNN, can also be conditioned on (observed) covariates that are modeled as additional conditioning variables in the latent space, which adds extra padding dimensions to the normalizing flow.

The other aspect of related works deals with multivariate probabilistic time series methods which are able to model high dimensional data. The Gaussian Copula Process method (Salinas et al., 2019a) is a RNN-based time series method with a Gaussian copula process output modeled using a low-rank covariance structure to reduce computational complexity and handle non-Gaussian marginal distributions. By using a low-rank approximation of the covariance matrix they obtain a computationally tractable method and are able to scale to multivariate dimensions in the thousands with state-of-the-art results. We will compare our model to this method in what follows.

## 4 TEMPORAL CONDITIONED NORMALIZING FLOWS

We denote the entities of a multivariate time series by $x_t^i \in \mathbb{R}$ for $i \in \{1, \ldots, D\}$ where $t$ is the time index. Thus the multivariate vector at time $t$ is given by $\mathbf{x}_t \in \mathbb{R}^D$. We will in what follows consider time series with $t \in [1, T]$, sampled from the complete time series history of our data, where for training we will split this time series by some context window $[1, t_0)$ and prediction window $[t_0, T]$.

In the `DeepAR` model (Salinas et al., 2019b), the log-likelihood of each entity $x_t^i$ at a time step $t \in [t_0, T]$ is maximized given an individual time series' prediction window. This is done with respect to the parameters of the chosen distributional model (e.g. negative binomal for count data) via the state of a RNN derived from its previous time step $x_{t-1}^i$ and current covariates $\mathbf{c}_t^i$. The emission distribution model, which is typically Gaussian for real-valued data or negative binomial for count data, is selected to best match the statistics of the time series and the network incorporates activation functions that satisfy the constraints of these distribution parameters, e.g. a `softplus()` for the scale parameter of the Gaussian.

A simple model for multivariate real-valued data could use a factorizing distribution in the emissions. Shared parameters can then learn patterns across the individual time series through the temporal component—but the model falls short of capturing dependencies in the emissions of the model. For this, a full joint distribution at each time step must be modeled, for example by using a multivariate Gaussian model. However, modeling the full covariance matrix not only increases the number of parameters of the neural network by $O(D^2)$, making learning difficult, but computing the loss becomes expensive when $D$ is large. Furthermore, statistical dependencies in the emissions would be limited to second-order effects. These models are referred to as `Vec-LSTM` in Salinas et al. (2019a).

We wish to have a scalable model of $D$ interacting time-series $\mathbf{x}_t$, and further to use a flexible distribution model on the emissions that allows for capturing and representing higher order moments. To this end, we model the conditional joint distribution at time $t$ of all time series $p_{\mathcal{X}}(\mathbf{x}_t | \mathbf{h}_t; \theta)$ with a flow, e.g. a Real NVP, conditioned on either the hidden state of a RNN at time $t$ or an embedding of the time series up to $t - 1$ from an attention module. In the case of an autoregressive RNN (either a LSTM or a GRU (Chung et al., 2014)), its hidden state $\mathbf{h}_t$ is updated given the previous time step observation $\mathbf{x}_{t-1}$ and the current time step's covariates $\mathbf{c}_t$ (as in Figure 1):

$$\mathbf{h}_t = \text{RNN}(\text{concat}(\mathbf{x}_{t-1}, \mathbf{c}_t), \mathbf{h}_{t-1}). \tag{3}$$

This model is autoregressive since it consumes the observation of the last time step $\mathbf{x}_{t-1}$ as well as the recurrent state $\mathbf{h}_{t-1}$ to produce the state $\mathbf{h}_t$ on which we condition the current observation.

To get a powerful and general emission distribution model, we stack $K$ layers of a conditional flow module (Real NVP or MAF) and together with the RNN, we arrive at our model of the conditional distribution of the future of all time series, given its past $t \in [1, t_0)$ and all the covariates in $t \in [1, T]$. As the model is autoregressive it can be written as a product of factors

$$p_{\mathcal{X}}(\mathbf{x}_{t_0:T} | \mathbf{x}_{1:t_0-1}, \mathbf{c}_{1:T}; \theta) = \Pi_{t=t_0}^T p_{\mathcal{X}}(\mathbf{x}_t | \mathbf{h}_t; \theta), \tag{4}$$

where $\theta$ denotes the set of all parameters of both the flow and the RNN.

For modeling the time evolution, we also investigate an encoder-decoder Transformer (Vaswani et al., 2017) architecture where the encoder embeds $\mathbf{x}_{1:t_0-1}$ and the decoder outputs the conditioning for the flow over $\mathbf{x}_{t_0:T}$ via a masked attention module. See Figure 2 for a schematic of the overall model in this case. While training, care has to be taken to prevent using information from future time points as well as to preserve the autoregressive property by utilizing a mask that reflects the causal direction of the progressing time, i.e. to mask out future time points. The Transformer allows the model to access any part of the historic time series regardless of temporal distance (Li et al., 2019) and thus is potentially able to generate better conditioning for the normalizing flow head.

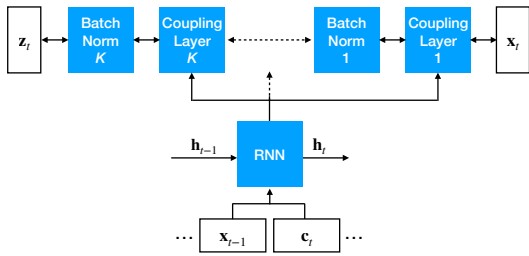

Figure 1: RNN Conditioned Real NVP model schematic at time $t$, consisting of $K$ blocks of coupling layers and Batch Normalization, where in each coupling layer we condition $\mathbf{x}_t$ and its transformations on the state of a shared RNN from the previous time step $\mathbf{x}_{t-1}$ and current time covariates $\mathbf{c}_t$ which are typically time dependent and time independent features.

In real-world data the magnitudes of different time series can vary drastically. To normalize scales, we divide each individual time series by their training window means before feeding it into the model. At inference the distributions are then correspondingly transformed with the same mean values to match the original scale. This rescaling technique simplifies the problem for the model, which is reflected in significantly improved empirical performance as noted in Salinas et al. (2019b).

## 4.1 TRAINING

Given $\mathcal{D}$, a batch of time series, where for each time series and each time step we have $\mathbf{x}_t \in \mathbb{R}^D$ and their associated covariates $\mathbf{c}_t$, we *maximize* the log-likelihood given by (1) and (3), i.e.

$$\mathcal{L} = \frac{1}{|\mathcal{D}|T} \sum_{\mathbf{x}_{1:T} \in \mathcal{D}} \sum_{t=1}^{T} \log p_{\mathcal{X}}(\mathbf{x}_t | \mathbf{h}_t; \theta)$$

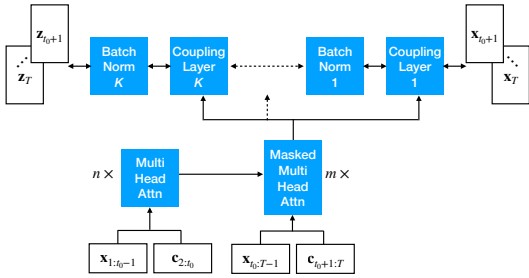

Figure 2: Transformer Conditioned Real NVP model schematic consisting of an encoder-decoder stack where the encoder takes in some context length of time series and then uses it to generate conditioning for the prediction length portion of the time series via a causally masked decoder stack. Note that the positional encodings are part of the covariates and unlike the RNN model, here all $\mathbf{x}_{1:T}$ time points are trained in parallel.

via SGD using Adam (Kingma & Ba, 2015) with respect to the parameters $\theta$ of the conditional flow and the RNN or Transformer. In practice, the time series $\mathbf{x}_{1:T}$ in a batch $\mathcal{D}$ are selected from a random time window of size $T$ within our training data, and the relative time steps are kept constant. This allows the model to learn to cold-start given only the covariates. This also increases the size of our training data when the training data has small time history and allows us to trade off computation time with memory consumption especially when $D$ or $T$ are large. Note that information about absolute time is only available to the RNN or Transformer via the covariates and not the relative position of $\mathbf{x}_t$ in the training data.

The Transformer has computational complexity $O(T^2D)$ compared to a RNN which is $O(TD^2)$, where $T$ is the time series length and the assumption that the dimension of the hidden states are proportional to the number of simultaneous time-series modeled. This means for large multivariate time series, i.e. $D > T$, the Transformer flow model has smaller computational complexity and unlike the RNN, all computation while training, over the time dimension happens in *parallel*.

## 4.2 COVARIATES

We employ embeddings for categorical features (Charrington, 2018), which allows for relationships within a category, or its context, to be captured while training models. Combining these embeddings

as features for time series forecasting yields powerful models like the first place winner of the Kaggle Taxi Trajectory Prediction[1] challenge (De Brébisson et al., 2015). The covariates $\mathbf{c}_t$ we use are composed of time-dependent (e.g. day of week, hour of day) and time-independent embeddings, if applicable, as well as lag features depending on the time frequency of the data set we are training on. All covariates are thus known for the time periods we wish to forecast.

### 4.3 INFERENCE

For inference we either obtain the hidden state $\hat{\mathbf{h}}_{t_1}$ by passing a "warm up" time series $\mathbf{x}_{1:t_1-1}$ through the RNN or use the cold-start hidden state, i.e. we set $\hat{\mathbf{h}}_{t_1} = \mathbf{h}_1 = \vec{0}$, and then by sampling a noise vector $\mathbf{z}_{t_1} \in \mathbb{R}^D$ from an isotropic Gaussian, go backward through the flow to obtain a sample of our time series for the next time step, $\hat{\mathbf{x}}_{t_1} = f^{-1}(\mathbf{z}_{t_1}|\hat{\mathbf{h}}_{t_1})$, conditioned on this starting state. We then use this sample and its covariates to obtain the next conditioning state $\hat{\mathbf{h}}_{t_1+1}$ via the RNN and repeat till our inference horizon. This process of sampling trajectories from some initial state can be repeated many times to obtain empirical quantiles of the uncertainty of our prediction for arbitrary long forecast horizons.

The attention model similarly uses a warm-up time series $\mathbf{x}_{1:t_1-1}$ and covariates and passes them through the encoder and then uses the decoder to output the conditioning for sampling from the flow. This sample is then used again in the decoder to iteratively sample the next conditioning state, similar to the inference procedure in seq-to-seq models.

Note that we do *not* sample from a reduced-temperature model, e.g. by scaling the variance of the isotropic Gaussian, unlike what is done in likelihood-based generative models (Parmar et al., 2018) to obtain higher quality samples.

## 5 EXPERIMENTS

Here we discuss a toy experiment for sanity-checking our model and evaluate probabilistic forecasting results on six real-world data sets with competitive baselines. The source code of the model, as well as other time series models, is available at https://github.com/zalandoresearch/pytorch-ts.

### 5.1 SIMULATED FLOW IN A SYSTEM OF PIPES

In this toy experiment, we check if the inductive bias of incorporating relations between time series is learnt in our model by simulating flow of a liquid in a system of pipes with valves. See Figure 3 for a depiction of the system.

Liquid flows from left to right, where pressure at the first sensor in the system is given by $S_0 = X + 3$, $X \sim \text{Gamma}(1, 0.2)$ in the shape/scale parameterization of the Gamma distribution. The valves are given by $V_1, V_2 \sim_{\text{iid}} \text{Beta}(0.5, 0.5)$, and we have

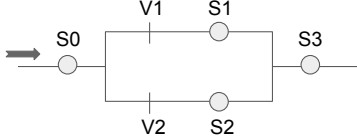

Figure 3: System of pipes with liquid flowing from left to right with sensors ($S_i$) and valves ($V_i$).

$$S_i = \frac{V_i}{V_1 + V_2} S_0 + \epsilon_i$$

for $i \in \{1, 2\}$ and finally $S_3 = S_1 + S_2 + \epsilon_3$ with $\epsilon_* \sim \mathcal{N}(0, 0.1)$. With this simulation we check whether our model captures correlations in space and time. The correlation between $S_1$ and $S_2$ results from both having the same source, measured by $S_0$. This is reflected by $\text{Cov}(S_1, S_2) > 0$, which is captured by our model as shown in Figure 4 left.

The cross-covariance structure between consecutive time points in the ground truth and as captured by our trained model is depicted in Figure 4 right. It reflects the true flow of liquid in the system from $S_0$ at time $t$ to $S_1$ and $S_2$ at time $t + 1$, on to $S_3$ at time $t + 2$.

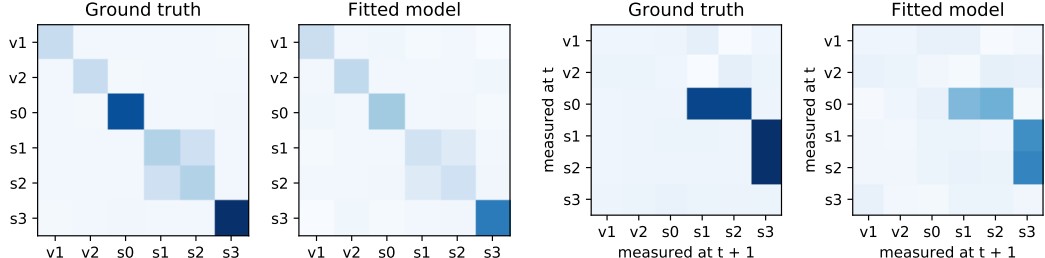

Figure 4: Estimated (cross-)covariance matrices. Darker means higher positive values. *left*: Covariance matrix for a fixed time point capturing the correlation between $S_1$ and $S_2$. *right*: Cross-covariance matrix between consecutive time points capturing true flow of liquid in the pipe system.

Table 1: Test set $\text{CRPS}_{\text{sum}}$ comparison (lower is better) of models from Salinas et al. (2019a) and our models `LSTM-Real-NVP`, `LSTM-MAF` and `Transformer-MAF`. The *two* best methods are in bold and the mean and standard errors of our methods are obtained by rerunning them 20 times.

| Data set | Vec-LSTM ind-scaling | Vec-LSTM lowrank-Copula | GP scaling | GP Copula | LSTM Real-NVP | LSTM MAF | Transformer MAF |
|---|---|---|---|---|---|---|---|
| Exchange | $0.008_{\pm 0.001}$ | $0.007_{\pm 0.000}$ | $0.009_{\pm 0.000}$ | $0.007_{\pm 0.000}$ | $\mathbf{0.0064}_{\pm 0.003}$ | $\mathbf{0.005}_{\pm 0.003}$ | $\mathbf{0.005}_{\pm 0.003}$ |
| Solar | $0.391_{\pm 0.017}$ | $0.319_{\pm 0.011}$ | $0.368_{\pm 0.012}$ | $0.337_{\pm 0.024}$ | $0.331_{\pm 0.02}$ | $\mathbf{0.315}_{\pm 0.023}$ | $\mathbf{0.301}_{\pm 0.014}$ |
| Electricity | $0.025_{\pm 0.001}$ | $0.064_{\pm 0.008}$ | $0.022_{\pm 0.000}$ | $0.024_{\pm 0.002}$ | $0.024_{\pm 0.001}$ | $\mathbf{0.0208}_{\pm 0.000}$ | $\mathbf{0.0207}_{\pm 0.000}$ |
| Traffic | $0.087_{\pm 0.041}$ | $0.103_{\pm 0.006}$ | $0.079_{\pm 0.000}$ | $0.078_{\pm 0.002}$ | $0.078_{\pm 0.001}$ | $\mathbf{0.069}_{\pm 0.002}$ | $\mathbf{0.056}_{\pm 0.001}$ |
| Taxi | $0.506_{\pm 0.005}$ | $0.326_{\pm 0.007}$ | $0.183_{\pm 0.395}$ | $0.208_{\pm 0.183}$ | $\mathbf{0.175}_{\pm 0.001}$ | $\mathbf{0.161}_{\pm 0.002}$ | $0.179_{\pm 0.002}$ |
| Wikipedia | $0.133_{\pm 0.002}$ | $0.241_{\pm 0.033}$ | $1.483_{\pm 1.034}$ | $0.086_{\pm 0.004}$ | $0.078_{\pm 0.001}$ | $\mathbf{0.067}_{\pm 0.001}$ | $\mathbf{0.063}_{\pm 0.003}$ |

## 5.2 REAL WORLD DATA SETS

For evaluation we compute the *Continuous Ranked Probability Score* (CRPS) (Matheson & Winkler, 1976) on each individual time series, as well as on the sum of all time series (the latter denoted by $\text{CRPS}_{\text{sum}}$). CRPS measures the compatibility of a cumulative distribution function $F$ with an observation $x$ as

$$\text{CRPS}(F, x) = \int_{\mathbb{R}} (F(z) - \mathbb{I}\{x \leq z\})^2 \, \mathrm{d}z \qquad (5)$$

where $\mathbb{I}\{x \leq z\}$ is the indicator function which is one if $x \leq z$ and zero otherwise. CRPS is a proper scoring function, hence CRPS attains its minimum when the predictive distribution $F$ and the data distribution are equal. Employing the empirical CDF of $F$, i.e. $\hat{F}(z) = \frac{1}{n} \sum_{i=1}^{n} \mathbb{I}\{X_i \leq z\}$ with $n$ samples $X_i \sim F$ as a natural approximation of the predictive CDF, CRPS can be directly computed from simulated samples of the conditional distribution (4) at each time point (Jordan et al., 2019). We take 100 samples to estimate the empirical CDF in practice. Finally, $\text{CRPS}_{\text{sum}}$ is obtained by first summing across the $D$ time-series—both for the ground-truth data, and sampled data (yielding $\hat{F}_{\text{sum}}(t)$ for each time point). The results are then averaged over the prediction horizon, i.e. formally

$$\text{CRPS}_{\text{sum}} = \mathbb{E}_t \left[ \text{CRPS}\left( \hat{F}_{\text{sum}}(t), \sum_i x_t^i \right) \right].$$

Our model is trained on the training split of each data set, and for testing we use a rolling windows prediction starting from the last point seen in the training data set and compare it to the test set. We train on `Exchange` (Lai et al., 2018), `Solar` (Lai et al., 2018), `Electricity`[2], `Traffic`[3], `Taxi`[4] and `Wikipedia`[5] open data sets, preprocessed exactly as in Salinas et al. (2019a), with their

---

[1]`https://www.kaggle.com/c/pkdd-15-predict-taxi-service-trajectory-i`

[2]`https://archive.ics.uci.edu/ml/datasets/ElectricityLoadDiagrams20112014`

[3]`https://archive.ics.uci.edu/ml/datasets/PEMS-SF`

[4]`https://www1.nyc.gov/site/tlc/about/tlc-trip-record-data.page`

[5]`https://github.com/mbohlkeschneider/gluon-ts/tree/mv_release/datasets`

properties listed in Table 2 of the appendix. Both `Taxi` and `Wikipedia` consist of count data and are thus dequantized before being fed to the flow (and mean-scaled).

We compare our method using LSTM and two different normalizing flows (`LSTM-Real-NVP` and `LSTM-MAF` based on Real NVP and MAF, respectively) as well as a Transformer model with MAF (`Transformer-MAF`), with the most *competitive* baseline probabilistic models from Salinas et al. (2019a) on the six data sets and report the results in Table 1. `Vec-LSTM-ind-scaling` outputs the parameters of an *independent* Gaussian distribution with mean-scaling, `Vec-LSTM-lowrank-Copula` parametrizes a low-rank plus diagonal covariance via Copula process. `GP-scaling` unrolls a LSTM with scaling on each individual time series before reconstructing the joint distribution via a low-rank Gaussian. Similarly, `GP-Copula` unrolls a LSTM on each individual time series and then the joint emission distribution is given by a low-rank plus diagonal covariance Gaussian copula.

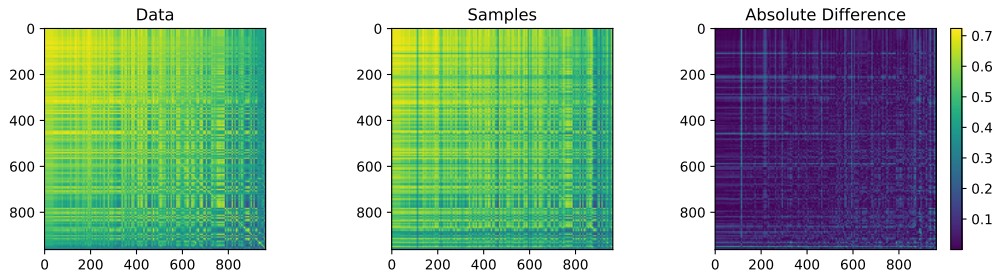

Figure 5: Visual analysis of the dependency structure extrapolation of the model. *Left*: Cross-covariance matrix computed from the test split of `Traffic` benchmark. *Middle*: Cross-covariance matrix computed from the mean of 100 sample trajectories drawn from the `Transformer-MAF` model's extrapolation into the future (test split). *Right*: The absolute difference of the two matrices mostly shows small deviations between ground-truth and extrapolation.

In Table 1 we observe that MAF with either RNN or self-attention mechanism for temporal conditioning achieves the state-of-the-art (to the best of our knowledge) $\text{CRPS}_{\text{sum}}$ on all benchmarks. Moreover, bipartite flows with RNN either also outperform or are found to be competitive w.r.t. the previous state-of-the-art results as listed in the first four columns of Table 1. Further analyses with other metrics (e.g. CRPS and MSE) are reported in Section B of the appendix.

To showcase how well our model captures dependencies in extrapolating the time series into the future versus real data, we plot in Figure 5 the cross-covariance matrix of observations (plotted left) as well as the mean of 100 sample trajectories (middle plot) drawn from `Transformer-MAF` model for the test split of `Traffic` data set. As can be seen, most of the covariance structure especially in the top-left region of highly correlated sensors is very well reflected in the samples drawn from the model.

## 6 CONCLUSION

We have presented a general method to model high-dimensional probabilistic multivariate time series by combining conditional normalizing flows with an autoregressive model, such as a recurrent neural network or an attention module. Autoregressive models have a long-standing reputation for working very well for time series forecasting, as they show good performance in extrapolation into the future. The flow model, on the other hand, does not assume any simple fixed distribution class, but instead can adapt to a broad range of high-dimensional data distributions. The combination hence combines the extrapolation power of the autoregressive model class with the density estimation flexibility of flows. Furthermore, it is computationally efficient, without the need of resorting to approximations (e.g. low-rank approximations of a covariance structure as in Gaussian copula methods) and is robust compared to Deep Kernel learning methods especially for large $D$. Analysis on six commonly used time series benchmarks establishes the new state-of-the-art performance against competitive methods.

A natural way to improve our method is to incorporate a better underlying flow model. For example, Table 1 shows that swapping the Real NVP flow with a MAF improved the performance, which is a

consequence of Real NVP lacking in density modeling performance compared to MAF. Likewise, we would expect other design choices of the flow model to improve performance, e.g. changes to the dequantization method, the specific affine coupling layer or more expressive conditioning, say via another Transformer. Recent improvements to flows, e.g. as proposed in the Flow++ (Ho et al., 2019), to obtain expressive bipartite flow models, or models to handle discrete categorical data (Tran et al., 2019), are left as future work to assess their usefulness. To our knowledge, it is however still an open problem how to model discrete ordinal data via flows—which would best capture the nature of some data sets (e.g. sales data).

### ACKNOWLEDGMENTS

K.R.: I would like to thank Rob Hyndman and Zaeem Burq for the helpful discussions and suggestions. I would like to acknowledge the traditional owners of the land on which I have lived and worked, the Wurundjeri people of the Kulin nation who have been custodians of their land for thousands of years. I pay my respects to their elders, past and present as well as past and present aboriginal elders of other communities.

We wish to acknowledge and thank the authors and contributors of the following open source libraries that were used in this work: GluonTS (Alexandrov et al., 2020), NumPy (Harris et al., 2020), Pandas (Pandas development team, 2020), Matplotlib (Hunter, 2007) and PyTorch (Paszke et al., 2019). We would also like to thank and acknowledge the hard work of the reviewers whose comments and suggestions have without a doubt help improve this paper.

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

## A    DATA SET DETAILS

Table 2: Properties of the data sets used in experiments.

| DATA SET | DIMENSION $D$ | DOMAIN | FREQ. | TOTAL TIME STEPS | PREDICTION LENGTH |
|---|---|---|---|---|---|
| EXCHANGE | 8 | $\mathbb{R}^+$ | DAILY | 6,071 | 30 |
| SOLAR | 137 | $\mathbb{R}^+$ | HOURLY | 7,009 | 24 |
| ELECTRICITY | 370 | $\mathbb{R}^+$ | HOURLY | 5,833 | 24 |
| TRAFFIC | 963 | $(0,1)$ | HOURLY | 4,001 | 24 |
| TAXI | 1,214 | $\mathbb{N}$ | 30-MIN | 1,488 | 24 |
| WIKIPEDIA | 2,000 | $\mathbb{N}$ | DAILY | 792 | 30 |

## B    ADDITIONAL METRICS

We used exactly the same open source code to evaluate our metrics as provided by the authors of Salinas et al. (2019a).

### B.1    COMPARISON AGAINST CLASSICAL BASELINES

We report test set $\text{CRPS}_{\text{sum}}$ results on `VAR` (Lütkepohl, 2007) a mutlivariate linear vector auto-regressive model with lags corresponding to the periodicity of the data, `VAR-Lasso` a Lasso regularized `VAR`, `GARCH` (van der Weide, 2002) a multivariate conditional heteroskedastic model, `GP` Gaussian process model, `KVAE` (Krishnan et al., 2017) a variational autoencoder on top of a linear state space model and `VES` a innovation state space model (Hyndman et al., 2008) in Table 3. Note that `VAR-Lasso`, `KVAE` and `VES` metrics are from (de Bézenac et al., 2020).

Table 3: Test set $\text{CRPS}_{\text{sum}}$ (lower is better) of classical methods and our `Transformer-MAF` model, where the mean and standard errors of our model are obtained over a mean of 20 runs.

| Data set | VAR | VAR-Lasso | GP | GARCH | VES | KVAE | Transformer MAF |
|---|---|---|---|---|---|---|---|
| Exchange | $0.010_{\pm 0.00}$ | $0.012_{\pm 0.000}$ | $0.011_{\pm 0.001}$ | $0.020_{\pm 0.000}$ | $\mathbf{0.005}_{\pm 0.00}$ | $0.014_{\pm 0.002}$ | $\mathbf{0.005}_{\pm 0.003}$ |
| Solar | $0.524_{\pm 0.001}$ | $0.51_{\pm 0.006}$ | $0.828_{\pm 0.01}$ | $0.869_{\pm 0.00}$ | $0.9_{\pm 0.003}$ | $0.34_{\pm 0.025}$ | $\mathbf{0.301}_{\pm 0.014}$ |
| Electricity | $0.031_{\pm 0.00}$ | $0.025_{\pm 0.00}$ | $0.947_{\pm 0.016}$ | $0.278_{\pm 0.00}$ | $0.88_{\pm 0.003}$ | $0.051_{\pm 0.019}$ | $\mathbf{0.0207}_{\pm 0.000}$ |
| Traffic | $0.144_{\pm 0.00}$ | $0.15_{\pm 0.002}$ | $2.198_{\pm 0.774}$ | $0.368_{\pm 0.00}$ | $0.35_{\pm 0.002}$ | $0.1_{\pm 0.005}$ | $\mathbf{0.056}_{\pm 0.001}$ |
| Taxi | $0.292_{\pm 0.00}$ | - | $0.425_{\pm 0.199}$ | - | - | - | $\mathbf{0.179}_{\pm 0.002}$ |
| Wikipedia | $3.4_{\pm 0.003}$ | $3.1_{\pm 0.004}$ | $0.933_{\pm 0.003}$ | - | - | $0.095_{\pm 0.012}$ | $\mathbf{0.063}_{\pm 0.003}$ |

## B.2 Continuous Ranked Probability Score (CRPS)

The average marginal CRPS over dimensions $D$ and over the predicted time steps compared to the test interval is given in Table 4.

Table 4: Test set CRPS comparison (lower is better) of models from Salinas et al. (2019a) and our models `LSTM-Real-NVP`, `LSTM-MAF` and `Transformer-MAF`. The mean and standard errors are obtained by re-running each method three times.

| Data set | Vec-LSTM ind-scaling | Vec-LSTM lowrank-Copula | GP scaling | GP Copula | LSTM Real-NVP | LSTM MAF | Transformer MAF |
|---|---|---|---|---|---|---|---|
| Exchange | $0.013_{\pm0.000}$ | $0.009_{\pm0.000}$ | $0.017_{\pm0.000}$ | $\mathbf{0.008}_{\pm\mathbf{0.000}}$ | $0.010_{\pm0.001}$ | $0.012_{\pm0.003}$ | $0.012_{\pm0.003}$ |
| Solar | $0.434_{\pm0.012}$ | $0.384_{\pm0.010}$ | $0.415_{\pm0.009}$ | $0.371_{\pm0.022}$ | $\mathbf{0.365}_{\pm\mathbf{0.02}}$ | $0.378_{\pm0.032}$ | $0.368_{\pm0.001}$ |
| Electricity | $0.059_{\pm0.001}$ | $0.084_{\pm0.006}$ | $0.053_{\pm0.000}$ | $0.056_{\pm0.002}$ | $0.059_{\pm0.001}$ | $\mathbf{0.051}_{\pm\mathbf{0.000}}$ | $0.052_{\pm0.000}$ |
| Traffic | $0.168_{\pm0.037}$ | $0.165_{\pm0.004}$ | $0.140_{\pm0.002}$ | $0.133_{\pm0.001}$ | $0.172_{\pm0.001}$ | $\mathbf{0.124}_{\pm\mathbf{0.002}}$ | $0.134_{\pm0.001}$ |
| Taxi | $0.586_{\pm0.004}$ | $0.416_{\pm0.004}$ | $0.346_{\pm0.348}$ | $0.360_{\pm0.201}$ | $0.327_{\pm0.001}$ | $\mathbf{0.314}_{\pm\mathbf{0.003}}$ | $0.377_{\pm0.002}$ |
| Wikipedia | $0.379_{\pm0.004}$ | $0.247_{\pm0.001}$ | $1.549_{\pm1.017}$ | $0.236_{\pm0.000}$ | $0.333_{\pm0.001}$ | $0.282_{\pm0.002}$ | $\mathbf{0.274}_{\pm\mathbf{0.007}}$ |

## B.3 Mean Squared Error (MSE)

The MSE is defined as the mean squared error over all the time series dimensions $D$ and over the whole prediction range with respect to the test data. Table 5 shows the MSE results for the the marginal MSE.

Table 5: Test set MSE comparison (lower is better) of models from Salinas et al. (2019a) and our models `LSTM-Real-NVP`, `LSTM-MAF` and `Transformer-MAF`.

| Data set | Vec-LSTM ind-scaling | Vec-LSTM lowrank-Copula | GP scaling | GP Copula | LSTM Real-NVP | LSTM MAF | Transformer MAF |
|---|---|---|---|---|---|---|---|
| Exchange | $\mathbf{1.6 \times 10^{-4}}$ | $1.9 \times 10^{-4}$ | $2.9 \times 10^{-4}$ | $1.7 \times 10^{-4}$ | $2.4 \times 10^{-4}$ | $3.8 \times 10^{-4}$ | $3.4 \times 10^{-4}$ |
| Solar | $9.3 \times 10^{2}$ | $2.9 \times 10^{3}$ | $1.1 \times 10^{3}$ | $9.8 \times 10^{2}$ | $\mathbf{9.1 \times 10^{2}}$ | $9.8 \times 10^{2}$ | $9.3 \times 10^{2}$ |
| Electricity | $2.1 \times 10^{5}$ | $5.5 \times 10^{6}$ | $\mathbf{1.8 \times 10^{5}}$ | $2.4 \times 10^{5}$ | $2.5 \times 10^{5}$ | $\mathbf{1.8 \times 10^{5}}$ | $2.0 \times 10^{5}$ |
| Traffic | $6.3 \times 10^{-4}$ | $1.5 \times 10^{-3}$ | $5.2 \times 10^{-4}$ | $6.9 \times 10^{-4}$ | $6.9 \times 10^{-4}$ | $\mathbf{4.9 \times 10^{-4}}$ | $5.0 \times 10^{-4}$ |
| Taxi | $7.3 \times 10^{1}$ | $5.1 \times 10^{1}$ | $2.7 \times 10^{1}$ | $3.1 \times 10^{1}$ | $2.6 \times 10^{1}$ | $\mathbf{2.4 \times 10^{1}}$ | $4.5 \times 10^{1}$ |
| Wikipedia | $7.2 \times 10^{7}$ | $3.8 \times 10^{7}$ | $5.5 \times 10^{7}$ | $4.0 \times 10^{7}$ | $4.7 \times 10^{7}$ | $3.8 \times 10^{7}$ | $\mathbf{3.1 \times 10^{7}}$ |

## C Univariate and Point Forecasts

Univariate methods typically give *better* forecasts than multivariate ones, which is counter-intuitive, the reason being the difficulty in estimating the cross-series correlations. The additional variance that multivariate methods add often ends up harming the forecast, even when one knows that individual time series are related. Thus as an additional sanity check, that this method is good enough to improve the forecast and not make it worse, we report the metrics with respect to a modern univariate point forecasting method as well as a multivariate point forecasting method for the `Traffic` data set.

Figure 6 reports the metrics from `LSTNet` (Lai et al., 2018) a *multivariate* point forecasting method and Figure 7 reports the metrics from `N-BEATS` (Oreshkin et al., 2020) a *univariate* model. As can be seen, our methods improve on the metrics for the `Traffic` data set and this pattern holds for other data sets in our experiments. As a visual comparison, we have also plotted the prediction intervals using our models in Figures 8, 9, 10 and 11.

## D Experiment Details

### D.1 Features

For hourly data sets we used hour of day, day of week, day of month features which are normalized. For daily data sets we use the day of week features. For data sets with minute granularity we use minute of hour, hour of day and day of week features. The normalized features are concatenated to

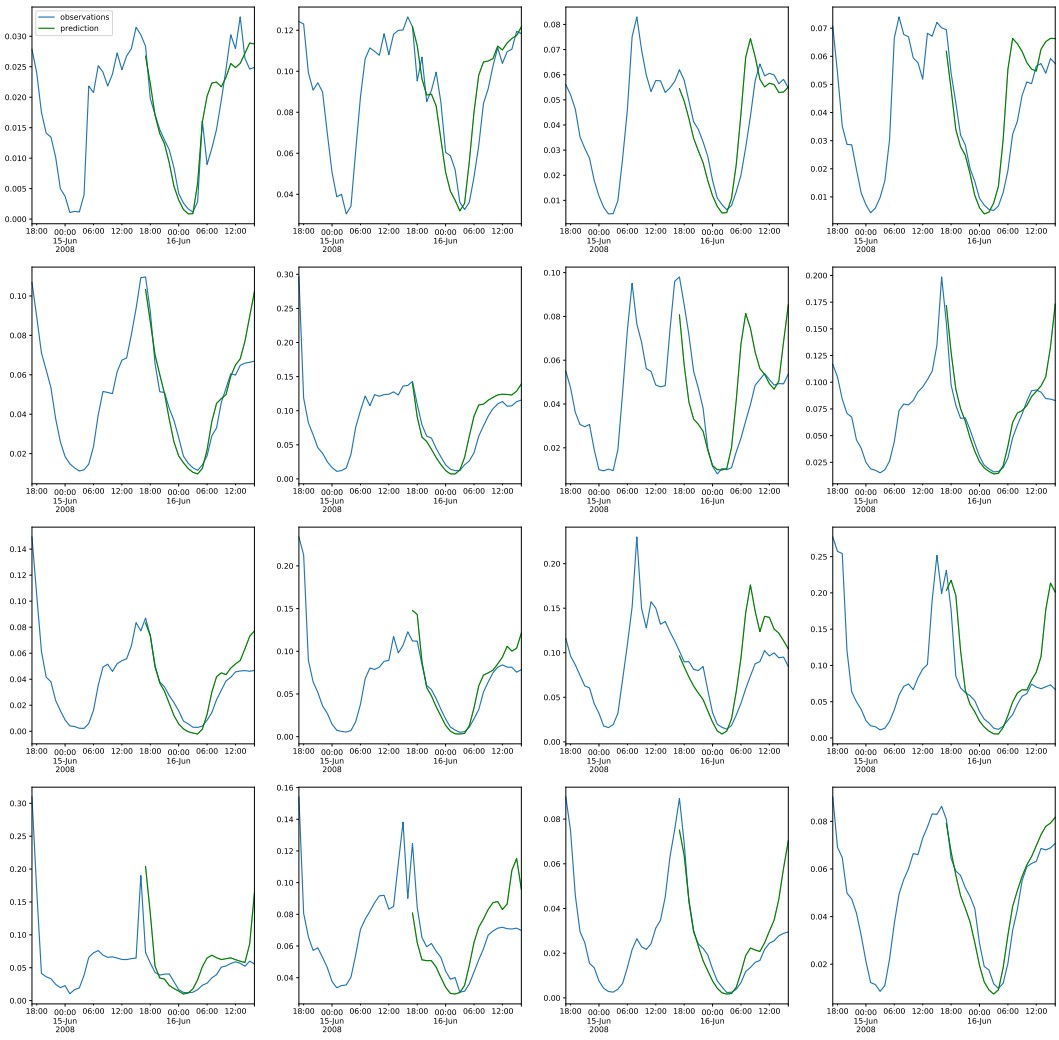

Figure 6: Point forecast and test set ground-truth from `LSTNet` multivariate model for `Traffic` data of the first 16 of 963 time series. $\text{CRPS}_{\text{sum}}$: 0.125, CRPS: 0.202 and MSE: $7.4 \times 10^{-4}$.

the RNN or Transformer input at each time step. We also concatenate lag values as inputs according to the data set's time frequency: $[1, 24, 168]$ for hourly data, $[1, 7, 14]$ for daily and $[1, 2, 4, 12, 24, 48]$ for the half-hourly data.

## D.2 HYPERPARAMETERS

We use batch sizes of $64$, with $100$ batches per epoch and train for a maximum of $40$ epochs with a learning rate of $1\text{e}{-3}$. The LSTM hyperparameters were the ones from Salinas et al. (2019a) and we used $K = 5$ stacks of normalizing flow bijections layers. The components of the normalizing flows ($f$ and $g$) are linear feed forward layers (with fixed input and final output sizes because we model bijections) with hidden dimensions of $100$ and ELU (Clevert et al., 2016) activation functions. We sample $100$ times to report the metrics on the test set. The Transformer uses $H = 8$ heads and $n = 3$ encoding and $m = 3$ decoding layers and a dropout rate of $0.1$. All experiments run on a single Nvidia V-100 GPU and the code to reproduce the results will be made available after the review process.

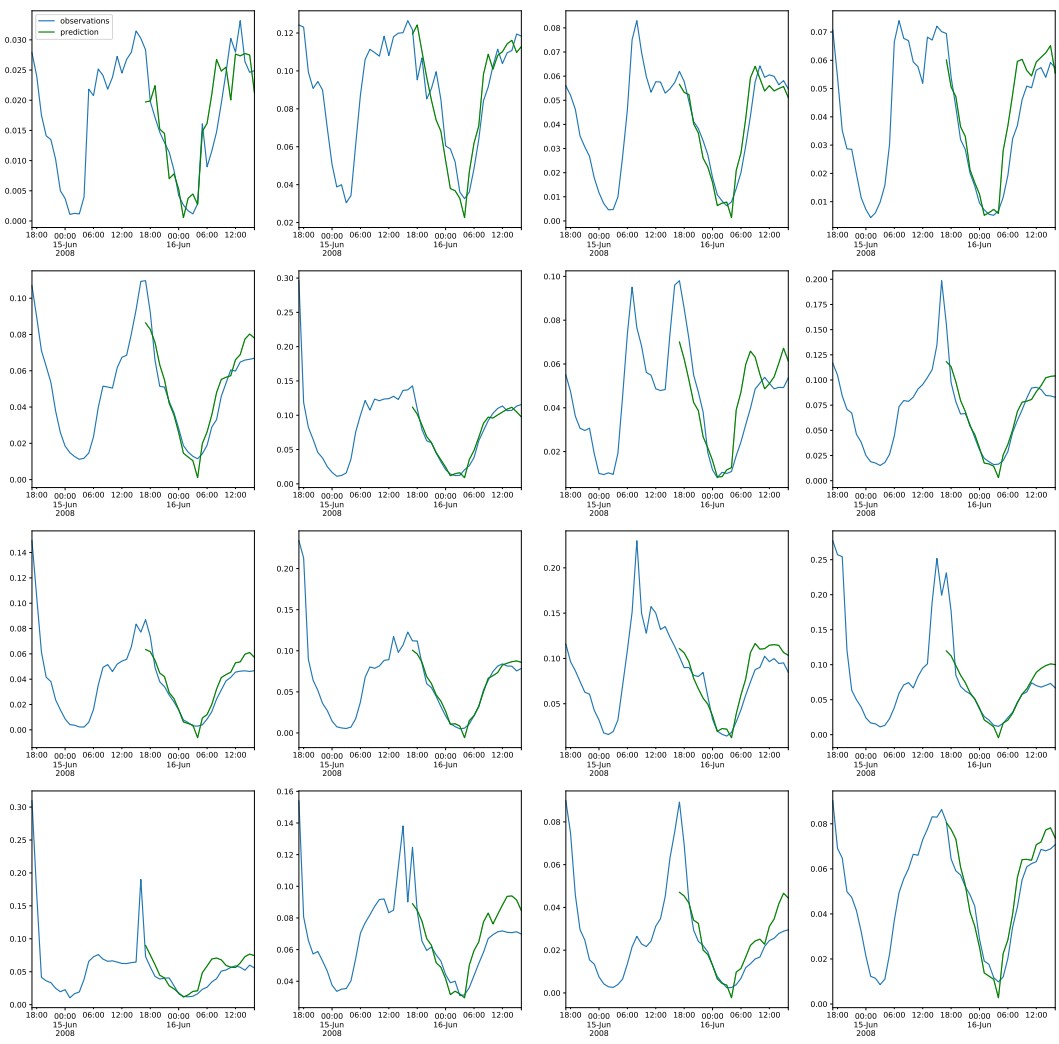

Figure 7: Point forecast and test set ground-truth from N-BEATS univariate model for Traffic data of the first 16 of 963 time series. CRPS$_{sum}$: 0.174, CRPS: 0.228 and MSE: $8.4 \times 10^{-4}$.

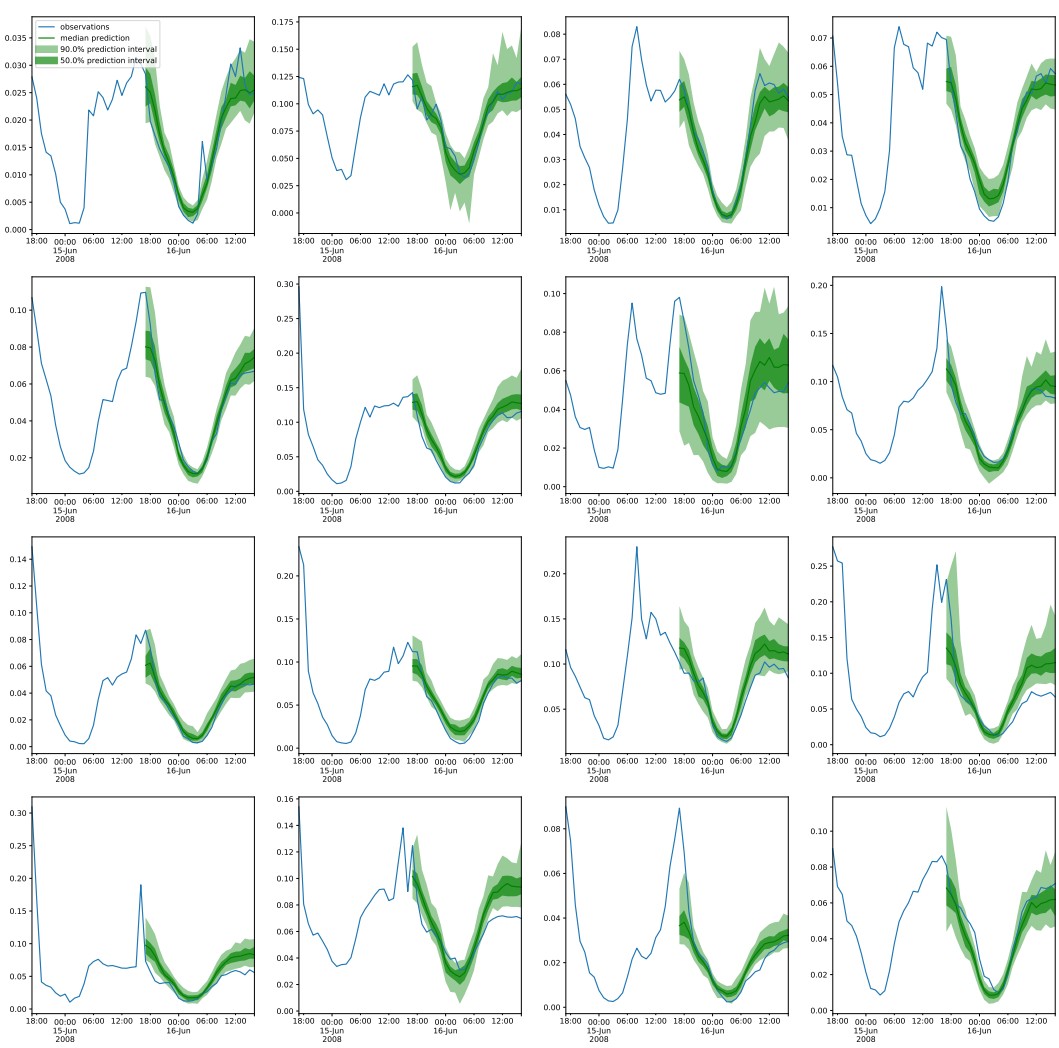

Figure 8: Prediction intervals and test set ground-truth from `LSTM-REAL-NVP` model for `Traffic` data of the first 16 of 963 time series.

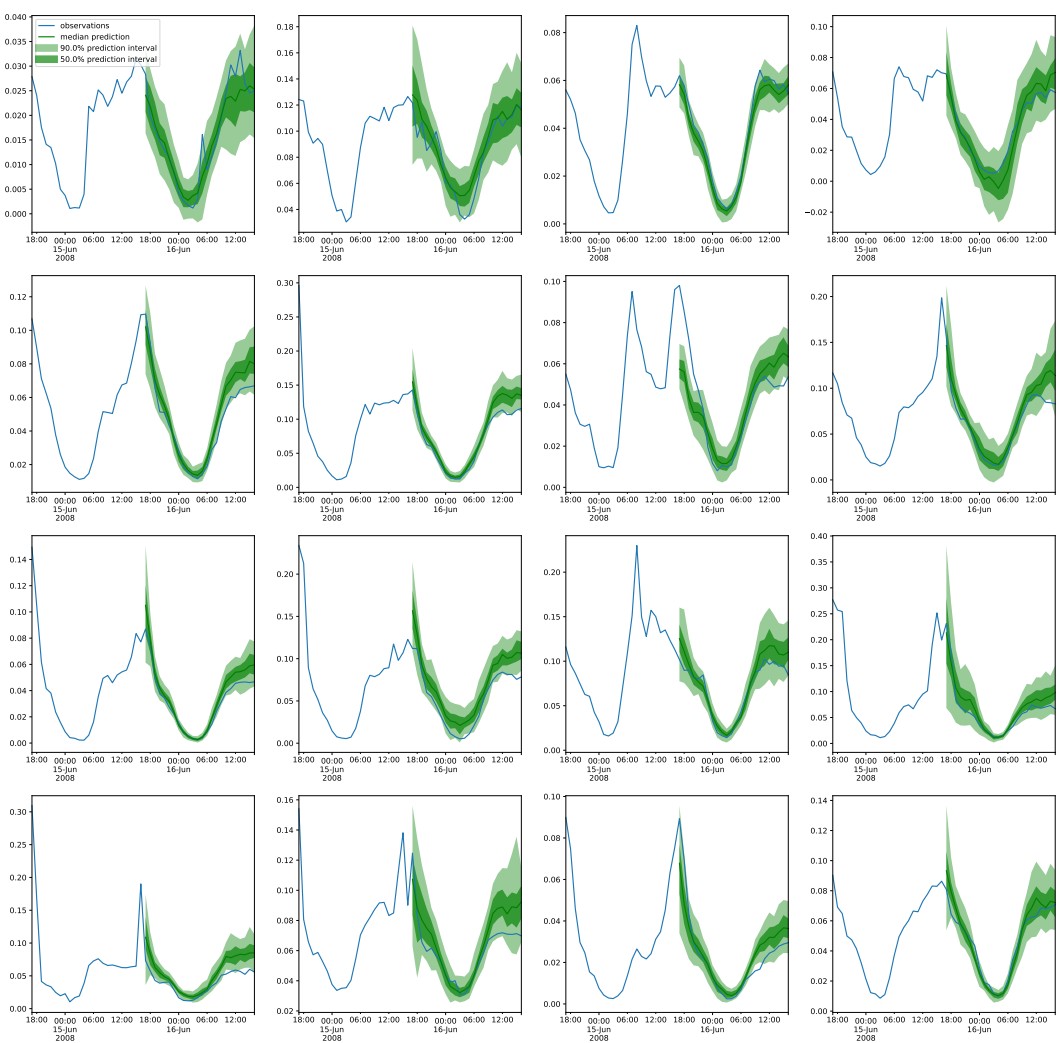

Figure 9: Prediction intervals and test set ground-truth from `Transformer-MAF` model for `Traffic` data of the first 16 of 963 time series.

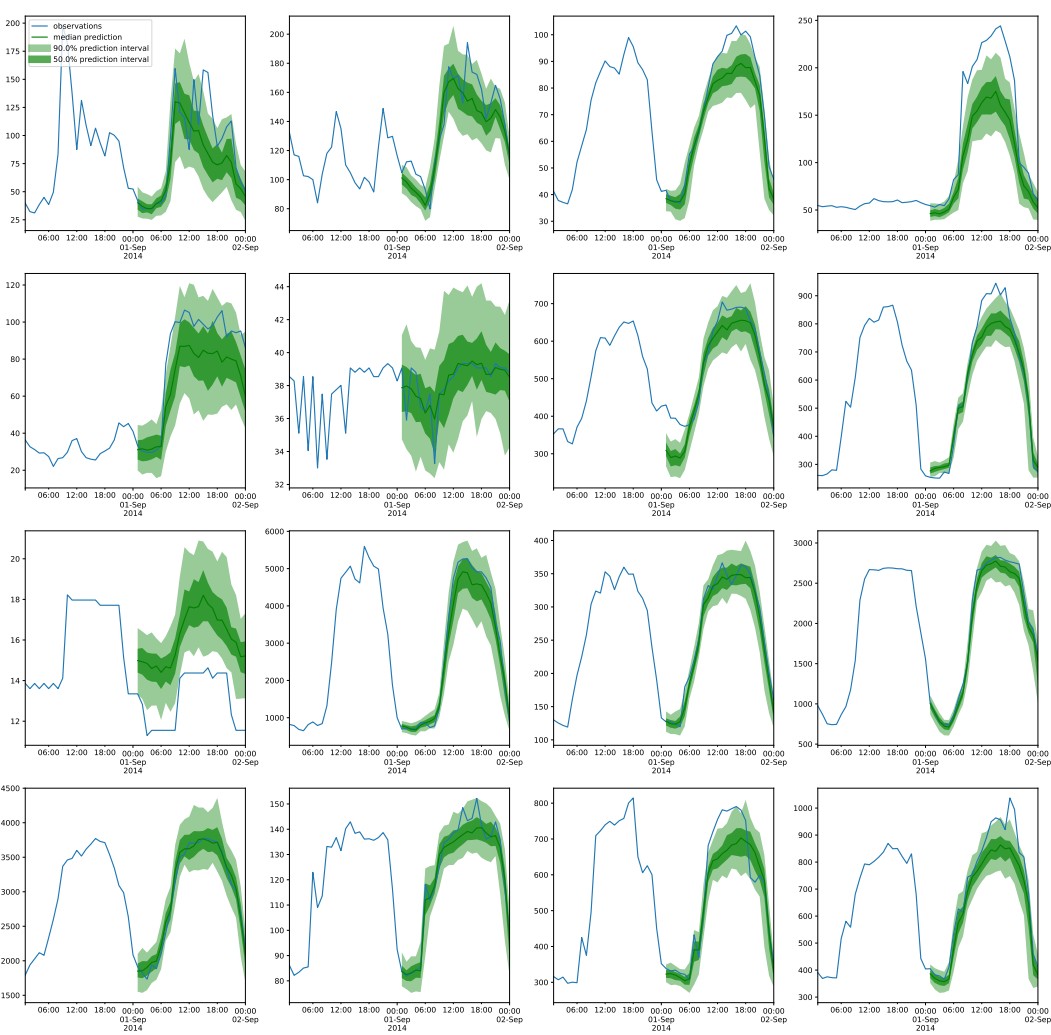

Figure 10: Prediction intervals and test set ground-truth from `LSTM-REAL-NVP` model for `Electricity` data of the first 16 of 370 time series.

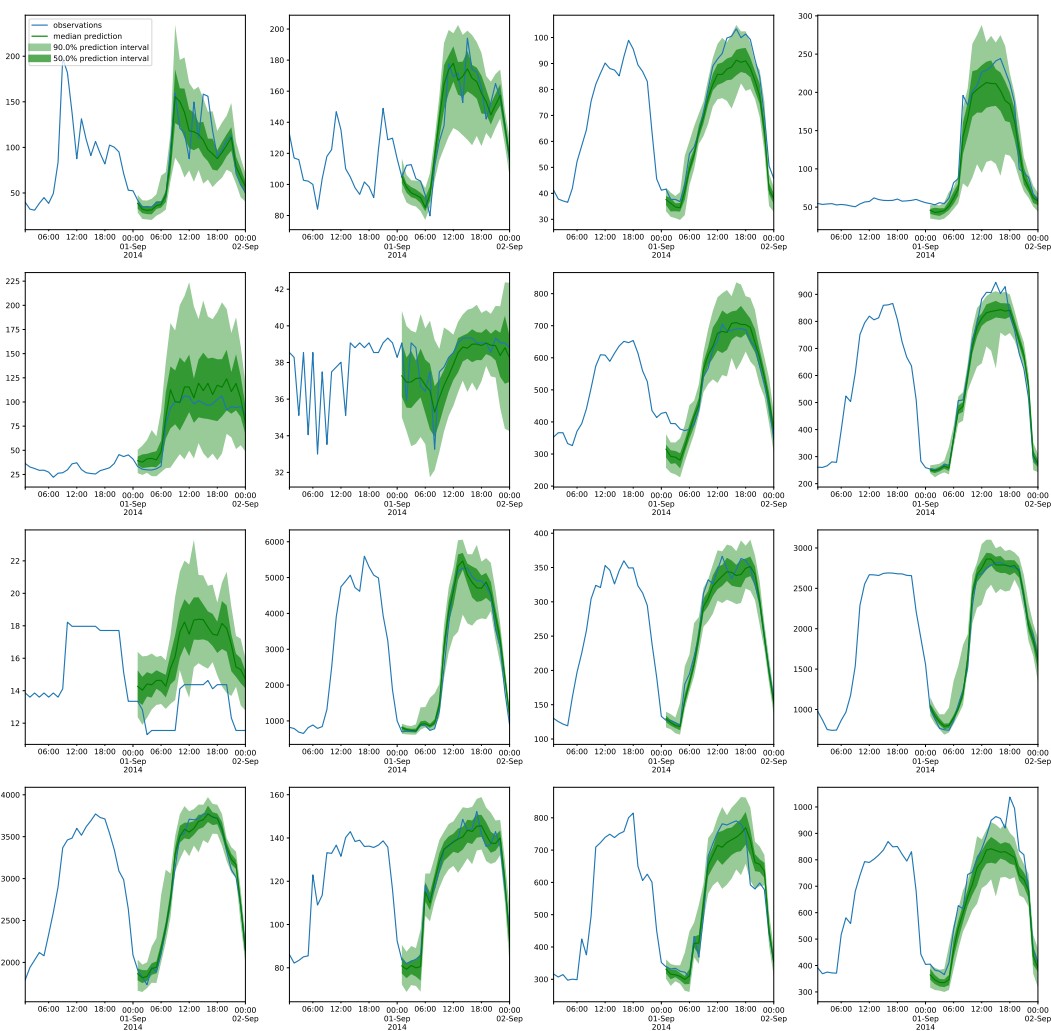

Figure 11: Prediction intervals and test set ground-truth from `Transformer-MAF` model for `Electricity` data of the first 16 of 370 time series.

