# OpenReview forum: "Multivariate Probabilistic Time Series Forecasting via Conditioned Normalizing Flows"
_ICLR.cc/2021/Conference — ICLR 2021 Spotlight_

### Official Review · AnonReviewer3 · 2020-10-27
**Recommendation to Accept**

**Rating:** 7
**Confidence:** 5

**Review:**

The paper proposes a method to provide probabilistic forecasts of multivariate time-series taking dependencies between series into account even for large dimensions. The approach consists in using a normalizing flow to model the distribution of observations at a time-step condition on a state that can be obtained either with a RNN or a transformer. The motivation of using the normalizing flow is to be able to model various type of distributions without having to make specific hypothesis on the distribution which could hinder accuracy. Experiments are performed both on a synthetic task and on real-world datasets where accuracy is shown to outperform previous methods.

### Strengths:
+ empirical improvements obtained over real-world datasets when measuring CRPS-sum accuracy metric for a problem that has significant applications
+ the clarity of the model description: models are well described and introduced. Generally speaking the paper is well written and reads well.

### Weaknesses:
- the improvement cannot be traced directly to the author contribution (the use of flow to model the multivariate distribution) given that they change the underlying model of the baseline (a LSTM) to a GRU. Given that the difference are small, it is unclear using GRU is not the source of improvement.
- the stability regarding the flow architecture is not analyzed whereas it is its core contribution. No detail is given on how the flow architecture was found, two architectures are used with 4 or 5 layers and no detail is given how they are chosen (the methods compared use only a single architecture on all datasets).
- the paper claims that one contribution is that their method is efficient but this is not backed-up by either theoretical complexity nor empirical data.
	- The time complexity given for RNN (O(TD^2)) is misleading given that this is a naive approach, for instance the GP copula methods have O(TD) runtime with a RNN.
	- The transformer method may still be faster due to parallelism but to make an efficiency claim, the runtime should be measured given that the theoretical complexity is higher

Despite the accuracy improvement and the fact that the paper is well written and motivated, I am currently not inclined to accept it given some ambiguities raised by the experiments but I believe those points could be clarified in the author rebuttal.

In particular, I think the following points needs clarification:
- running their method with an LSTM would be very important to have an apple-to-apple comparison and discard that the improvement comes from GRU rather than normalizing flows
- How did you find K = 4 or K = 5 parameters? by looking at a single dataset? How do you choose between 4 and 5 in your test-set? Given that the methods you compare with use a single configuration, the comparison should also be made with a single configuration.
- a performance claim is made but is not backed-up at the moment. The complexity of the models are O(TD^2) or O(T^2D) but the GP copula baseline is O(TD). Rather than complexity, perhaps runtime would be a better advocate for this point? (otherwise I believe the efficiency claim should be dropped as it is not backed up by theoretical complexity nor runtime measurement)

# Additional feedback (no part of the decision)

Details:
* section 4: "In the case of an autoregressive".
* the artificial experiments is ill-suited to show-case the benefit of the method, couldnt one model directly a full-covariate matrix in this case given the low-dimensionality? How many time-steps were considered for the sequence of pipes?

Finally, I would like to note that I reviewed a previous version of this paper and that I updated my review with the changes made by the author.

---

### Official Review · AnonReviewer4 · 2020-10-27
**A good idea but may need more work**

**Rating:** 6
**Confidence:** 3

**Review:**

This work explores combining an RNN and a neural density estimator for forecasting in multivariate time series. RNN is stacked with a density estimator, MAF for best results, to forecast density of a multivariate time series at future time steps. In addition, variations of the architecture with attention and other density estimators are examined. The architecture, RNN+MAF and variants, is evaluated by CRPS score on several datasets.

The work may have  a value but I have many issues reading it. I have a feeling that the motivation and the  related work are incomplete. Too many variants of the architectures are discussed and unnecessary details are provided while important questions are not discussed.  Reconstructing the 'best' architecture from the paper is not a trivial task. Empirical evaluation is somewhat sketchy.

In detail,

* while deep learning is a powerful tool for forecasting in multivariate timeseries, including settings with uncertainty, it is not the only way. 'Classical' approaches like Kalman filter and multi-task gaussian processes work well in many settings.

* there was work on forecasting time series with uncertainty using end-to-end deep learning architectures (rather than involving GP GCP etc) which the paper does not cite. For example, https://deepai.org/publication/sequential-neural-models-with-stochastic-layers and other variants of stochastic RNNs, as well as neural processes, https://arxiv.org/abs/1807.01622.

* As I read the paper, RNN+MAF gives the best results empirically, and MAF is indeed a sato density estimator, so this is quite expected. However, if the purpose is *forecasting* that is, obtaining the future states, I am wondering why isn't IAF used instead, which is more suited for sampling.

* Empirical evaluation --- on the toy example, I am wondering why the learned cross-variance matrices are visually different in Figure 4. I would expect them to be visually indistinguishable on such a simple model. Even using simpler multi-variate time series models. In Table 1, "the mean and standard errors are obtained by re-running ... three times". I see that the same method of "obtaining the mean and standard error" appears in the cited work, but it does not make the method more credible. You cannot "obtain" or even reliably estimate the mean and the standard error from three runs. Either a statistically credible number of runs is needed, or there isn't much sense in reporting the standard error.

~~~~~

Based on the discussion and the updated manuscript, I am happy to say that most my concerns were addressed and at least partially resolved.  I have updated my score accordingly.

---

### Official Review · AnonReviewer2 · 2020-10-28
**Useful contribution to the field of probabilistic time series forecasting**

**Rating:** 9
**Confidence:** 5

**Review:**

The paper presents a new approach to multivariate probabilistic time series forecasting. The authors propose to combine recurrent neural networks with conditioned normalizing flows to model the output distribution. They explain a few variations of the method and evaluate them against the baselines.

*Quality*
The choice of the output distribution is always a hard step in the probabilistic time series forecasting using RNNs. The authors propose an interesting workaround by trying to learn the shape of the output distribution directly with conditioned normalizing flows. In combination with a Transformer model, this also has the potential to reduce training/inference time for high-dimensional datasets. The authors explain and motivate the proposed method really well and I think the paper is a useful contribution to the field of probabilistic time series forecasting.

*Clarity*
The paper is very well-written.

*Originality*
To my best knowledge, the proposed approach is new.

*Significance*
Rather significant. The multivariate forecasting models suffer from the problems with scaling and the paper’s proposed combination of transformers with conditioned normalizing flows has the potential to overcome this problem.

Pros
* Conditioned normalizing flows can ease the design choice of the output distribution
* The method scales well in terms of number of time series to forecast
* Great empirical results

Cons
* Since the scaling is a big issue for multivariate models, it would be useful to compare training/inference times between the evaluated methods.

---

### Official Review · AnonReviewer1 · 2020-11-02
**Interesting model, but falls a bit short on the experimental evaluation**

**Rating:** 7
**Confidence:** 4

**Review:**

# Update after author discussion
My original review mentioned an apparent similarity to VideoFlow and a weak experimental setup as the main reasons for my score. Through discussions with the authors, I am now convinced that VideoFlow is indeed a quite different model, which significantly adds to the novelty of the proposed model.

As for the experiments, I had misunderstood the setup. In terms of probabilistic multivariate methods, my experience lies in the multi-output Gaussian process domain where it is common to consider tasks where only some outputs are considered missing (i.e., part of the test set), thus testing the models' ability to make use of partial information at a given input. This was the setup I had in mind, but the proposed model is not easily adapted for this. This is not a limitation of the proposed model as much as an inherent difficulty of handling missing data with neural networks. The experimental setup used in this paper instead considers a training set and a test set entirely separated in time, which is not a setup where GPs usually perform well, so the additional experiments I had requested do not make much sense. Indeed, the baseline methods chosen for comparison in the paper are both reasonable and strong.

In conclusion, the authors have satisfactorily addressed my concerns with the submission. I still think the proposed idea is good (although not groundbreakingly novel), the paper is well-written, and the model thoroughly discussed and tested. I view the submission as a significant contribution to time-series modelling and is therefore recommending acceptance.


----------------------------------------------------------


# Summary
The paper proposes to use normalising flows for probabilistic modelling of multivariate time-series. By taking advantage of normalising flows' ability to model complex interactions between a large number of outputs with arbitrary output distributions, the proposed model naturally scales to many concurrent time-series. The temporal component is modelled with an autoregressive model, such as an RNN or a Transformer, which outputs a latent representation of previous times and covariates, which is then used as the condition in the flow transformations.
Combined, the two parts of the model allow for probabilistic trajectories to be sampled for predictions into the future. The model is tested on six datasets and compared to four other models, showing very good performance.

# Evaluation
## Strong points
* Using normalising flows for multivariate time-series modelling is a good idea.
* Two variations of the proposed model are tested, showing how design decision change the performance.

## Weak points
* Limited novelty over previous models.
* Experimental section a bit weak.

## Recommendation
Borderline reject.

The model in itself is interesting, as is the application to time-series modelling. However, the model does seem very related to VideoFlow, which is mentioned briefly in the paper but never discussed properly, so the novelty seems limited. Furthermore, the experimental section only considers competing models from one other paper.

## Detailed feedback
The paper is well-written, well-structured, and the background section is comprehensive and detailed. The writing is clear and of high quality.

The related work section discusses some relevant work, but given that the paper concerns probabilistic time-series modelling, I would have expected to see some discussions of multi-output Gaussian processes (MO-GPs) and state-space models. I am also quite surprised how little discussion there is of Kumar et al. (2019) given that VideoFlow is very related to the proposed model. The way the two models introduce temporal dependencies differ, but they are otherwise similar. A comparison of the two models seems in order.

The proposed model is certainly interesting, and the idea of using flows to correlate time-series is very good. Again, it is not entirely new, but it is a very underexplored path. There is not much novelty from a theoretical perspective either, as the model is mostly a combination of two sub-models; a normalising flow architecture and an autoregressive component. I see this as a strength of a kind, as it makes the model quite modular, which should make it interesting to practitioners - it's just that there is a trade-off in novelty.

To me, the main shortcoming of the paper is experimental evaluation. The model is evaluated on six datasets, which are quite different, so this is definitely good. The experiments are run several times and results are reported with standard deviations - excellent! The model, however, is compared against two baselines and two competing models, which seems thorough, but the two baselines are very related to each other, as are the two competing models. All four models are taken from the same paper (Salinas et al. 2019a). Given how similar this model is to VideoFlow, I do not understand why that model was not tested. Furthermore, I think it would have made sense to include independent Gaussian processes (GPs) as baselines and to test additional multi-output GP models, e.g., a semi-parametric latent factor model and the Gaussian process autoregressive regression model (GPAR, Requeima et al., 2019).
In addition, I find the CRPS quite hard to interpret, and I think it would have made sense to include more metrics than just those based on CRPS. MAE is included in the supplementary, but reporting something like the standardised MSE and the (negative) log probability of the data under the model would have been useful too. In particular, the log probability would have been interesting to see.

Lastly, I think it would be interesting to see plots of the predictive densities by the different models for a couple of datasets. This is, of course, only a qualitative assessment, but I am very interested in seeing how the probabilistic trajectory of the model looks like in terms of smoothness and predictive density.

# Questions
Comments that will affect my score:
1. Could you please include experiments with other models? In particular, I'd be interested in seeing the performance of VideoFlow and GPAR.
2. Could you please include additional metrics? In particular, I'd be interested in seeing the negative predictive log probability.
3. Could you please include some plots of the predictions of the models on future data so we can assess the quality of the predictions? (Including these in the supplementary would be fine.)

Minor questions not likely to affect my score:
1. You say that "Shared parameters can then learn patterns across the individual time series through the temporal component - but the model falls short of capturing dependencies in the emissions of the model." Why can't shared parameters capture dependencies between outputs?
2. In section 4.3 you mention a cold start, but I am confused about when you would ever prefer this to a warm-start. Can you elaborate?

---

### Decision · Program_Chairs · 2021-01-07
**Final Decision**

**Decision:**

Accept (Spotlight)

**Comment:**

This paper proposes an approach to probabilistic time series forecasting based on combining autoregressive deep learning models with normalizing flows. In terms of strengths, time series forecasting is a fundamental problem. The proposed approach is a reasonable combination of existing model components that provides a flexible, end-to-end trainable framework for multivariate probabilistic forecasting. The experiments are well-conducted and the results outperform recently published methods. While the reviewers raised a number of questions, all of the reviewers agree that their questions have be answered satisfactorily by the authors during the discussion and the paper should be accepted. The authors should be sure to incorporate the reviewer suggestions and author responses into the final paper.